# Weak Fault Feature Extraction Method Based on Improved Stochastic Resonance

**DOI:** 10.3390/s22176644

**Published:** 2022-09-02

**Authors:** Zhen Yang, Zhiqian Li, Fengxing Zhou, Yajie Ma, Baokang Yan

**Affiliations:** 1Engineering Research Center for Metallurgical Automation and Measurement Technology of Ministry of Education, Wuhan University of Science and Technology, Wuhan 430081, China; 2School of Artificial Intelligence, Wuchang University of Technology, Wuhan 430223, China

**Keywords:** stochastic resonance, early weak fault, feature extraction, comprehensive evaluation index, parameter optimization

## Abstract

Aiming at the problems of early weak fault feature extraction of bearings in rotating machinery, an improved stochastic resonance (SR) is proposed combined with the advantage of SR to enhance weak characteristic signals with noise energy. Firstly, according to the characteristics of the large parameters of the actual fault signal, the amplitude transform coefficient and frequency transform coefficient are introduced to convert the large parameter signal into small parameter signal which can be processed by SR, and the relationship of second-order parameters are introduced. Secondly, a comprehensive evaluation index (CEI) consisted of power spectrum kurtosis, correlation coefficient, structural similarity, root mean square error, and approximate entropy, is constructed through BP neural network. Moreover, this CEI is adopted as fitness function to search the optimal damping coefficient and amplitude transform coefficient with adaptive weight particle swarm optimization (PSO). Finally, according to the improved optimal SR system, the weak fault feature can be extracted. The simulation and experiment verify the effectiveness of the proposed method compared with traditional second-order general scale transform adaptive SR.

## 1. Introduction

Status monitoring and fault diagnosis of rotating mechanical components such as bearings, gears, and rotors are significant for ensuring the safe operation of equipment. When the components fail, impulses reflecting their structural defects can be found in vibration signals or acoustic signals [1]. Actually, early fault signals are always submerged by serious background noise. Traditional signal filtering methods usually adopt noise suppression strategy to enhance fault signals, but they would also suppress fault signals when suppress noise, which means that it is difficult to extract weak fault features effectively. Stochastic Resonance can use the energy of noise to strengthen the original weak fault signal through nonlinear system, which can improve the signal-to-noise ratio (*SNR*) of the system output, and it is widely used in the field of fault diagnosis [2].

In 1981,Benzi proposed the concept of SR when he studied the periodic changes in the “glacial period” and “warm period” of the earth’s climate [3]. After decades of development, SR has been proved as an effective method for detecting weak signals. However, constrained by the adiabatic approximation theory and the linear response theory, the classical SR theory is limited by small parameters; that is, the signal frequency and signal amplitude must be far less than 1, but the actual engineering signals often cannot meet the requirements [4]. Therefore, a scale transformation SR theory is proposed to achieve scale transformation by compressing the target signal frequency or decomposing the target signal. Tang proposed frequency-shifted and re-scaling stochastic resonance (FRSR) to realize the detection of large signals, which used the dual means of frequency shifting and frequency scaling to compress the signal frequency [5]. Wang proposed multi-scale noise tuning stochastic resonance (MSTSR) to identify the characteristic frequency of bearings in which the signal is decomposed and reconstructed through wavelet packet transform [6]. Leng proposed twice sampling stochastic resonance (TSSR) to detect weak signal overwhelmed in noise under large parameters conditions [7]. Kong proposed normalized scale transformation stochastic resonance (NSTSR) to transform the large parameter signal into small parameter signal by introducing the frequency compression parameters [8]. The above scale transform methods only use one scale transform coefficient to satisfy the small parameter condition. Methods of SR with single scale transform coefficient only consider the frequency matching but ignore the relationship between the signal amplitude and the threshold amplitude of SR system, which would result in an inability to achieve the best stochastic resonance effect.

SR describes the optimal matching between signal, noise and nonlinear system [9]. For a certain fault signal, the useful signal and noise are unchanged. Therefore, adjusting the parameters of nonlinear system is a significant way to achieve the optimal matching. How to obtain the optimal parameters of SR system is a research hotspot. Zhang proposed an adaptive SR method based on the grey wolf optimizer to diagnose the fault of rolling bearing and gearbox in which *SNR* is used as an evaluation index [10]. Lei introduced an adaptive multi-stable SR method based on quantum genetic algorithm to extract the early fault feature of bearing in which the weighted signal to noise ratio (WSNR) is used as the fitness function [11]. He proposed a power function type bistable SR method to detect fault signal, which combined the power function single potential well model with the Gaussian Potential model and used the average signal to noise ratio gain as the measurement index [12]. These methods all use the *SNR* or its deformation as the evaluation index, but when the target signal frequency cannot be accurately determined, *SNR* index cannot be used as the evaluation index for the optimization of the parameters of the SR system.

Aiming at the difficulty of single scale transform coefficient to match the signal amplitude and characteristic frequency at the same time, a second-order amplitude-frequency re-scaling match (SAFRM) SR method is proposed, which introduces the amplitude transform coefficient and frequency transform coefficient to realize the optimal match of signal, noise and nonlinear system. Aiming at the difficult of the *SNR* calculation in engineering signal, a new comprehensive evaluation index (CEI) is proposed, which uses the BP neural network to fuse five indexes of power spectrum kurtosis, correlation coefficient, structural similarity, root mean square error and approximate entropy. This CEI can overcome the reliance on unknown characteristic frequency, and the SR system can obtain the optimal parameters when CEI obtains the minimum value. So, through the CEI-based adaptive weight particle swarm optimization (APSO) algorithm, the optimal parameter values of SR system can be obtained, thus, through this optimal SR system, weak fault characteristic signal can be extracted.

## 2. Basic Theory

### 2.1. SR Theory Analysis

A second-order bistable system subjected to noise and external periodic driving force can be described by the following Langevin equation:(1)$$\begin{array}{l} \frac{d^{2}x(t)}{dt^{2}}+\gamma \frac{dx(t)}{dt}=-\frac{dU(x)}{dx}+S(t)+N(t)\\ U(x)=-\frac{a}{2}x^{2}+\frac{b}{4}x^{4}\begin{array}{cccc} , & & & a,b>0 \end{array}\\ S(t)=A\cos(2\pi f_{m}t+\varphi )\\ N(t)=\sqrt{2D}\xi (t) \end{array}$$
where x(t) is the output signal, γ is the damping factor; U(x) is the bistable potential function; S(t) is the periodic signal and the amplitude is A, the frequency is $f_{m}$, the phase is φ; N(t) is the noise, and D is the noise intensity, ξ(t) is Gaussian white noise with zero-mean and unit-variance [13,14,15].

The output signal x(t) can be understood as the movement trajectory of the unit mass Brownian particle in the potential field U(x) under the combined action of the damping force −γdx(t)/dt, potential field force −dU(x)/dx, periodic driving force S(t) and random noise N(t). The potential function U(x) has three equilibrium points, stable equilibrium point $\pm x_{m}$ and an unstable equilibrium point $x_{0}$, which take the minimum value at $x_{m}=\pm \sqrt{a/b}$ and the maximum value at $x_{0}=0$, it generates a symmetrical double potential separated by a potential barrier with barrier height of $\Delta U=a^{2}/(4b)$. The bistable potential function U(x) (a=b=1) is illustrated in Figure 1.

When the excitation signal is a noise-free periodic signal S(t), the bistable potential changes periodically according to the frequency $f_{m}$ driven by the excitation signal. At this time, there is a threshold amplitude $A_{c}$ in the system, and $A_{c}=\sqrt{4a^{3}/(27b)}$. When $A<A_{c}$, the Brownian particle would be hardly to jump over the potential barrier, it would be only to oscillate in one side of potential well; when $A>A_{c}$, the Brownian particle would jump over the potential barrier and make regular transition movement between double potential wells.

The bistable system described by the second order Equation (1) is transformed into the equivalent system of two first order differential Equation (2), where dx/dt is the time derivative.
(2)$$\begin{array}{l} \frac{dx}{dt}=y\\ \frac{dy}{dt}=-\frac{dU}{dx}-\gamma y+S(t)+N(t) \end{array}$$

When the excitation signal is pure noise N(t), the transition probability density response ρ=ρ(x,y,t) can be described by the corresponding Fokker-Planck equation, in the form of a Boltzmann equation [16].
(3)$$\frac{\partial \rho }{\partial t}+y\frac{\partial \rho }{\partial x}-\frac{\partial U(x)}{\partial x}\frac{\partial \rho }{\partial y}=\gamma \frac{\partial }{\partial y}y\rho +D\frac{\partial^{2}\rho }{\partial y^{2}}$$

According to the three points $\left(x_{0},y_{0})=(0,0\right)$, $\left(x_{-},y_{-})=(-\sqrt{a/b},0\right)$ and $\left(x_{+},y_{+})=(\sqrt{a/b},0\right)$ of the bistable system, the probability distribution function of the bistable system during the whole “quasi-stable” period is shown in Equation (4):(4)$$\rho (x,y,t)=\left\{ \begin{array}{l} \rho_{+}(x,y,t)=N_{+}\exp[-\frac{\tilde{U}(x,y,t)}{D}],\begin{array}{cc} & x>x_{0} \end{array}\\ \rho_{-}(x,y,t)=N_{-}\exp[-\frac{\tilde{U}(x,y,t)}{D}],\begin{array}{cc} & x\leq x_{0} \end{array} \end{array}\right.$$
where $N_{+}$ and $N_{-}$ represents the normalization constant, $\tilde{U}(x,y,t)$ is the generalized potential function that can be obtained by utilizing the small parameter expansion method as:(5)$$\tilde{U}(x,y,t)=\frac{1}{2}y^{2}-\frac{a}{2}x^{2}+\frac{b}{4}x^{4}$$

From Equation (5), the Kramers escape rates $R_{-}$ and $R_{+}$ from the $\left(x_{-},y_{-}\right)$ and $\left(x_{+},y_{+}\right)$ potential wells can be obtained as:(6)$$\begin{array}{l} R_{-}=\frac{1}{2\pi }\sqrt{\left|\tilde{U^{\prime \prime }}(x_{-},y_{-},t)\tilde{U^{\prime \prime }}(x_{0},y_{0},t)\right|}\exp(\frac{\tilde{U}(x_{-},y_{-},t)-\tilde{U}(x_{0},y_{0},t)}{D})=\frac{a}{2\sqrt{2}\pi \gamma }\exp(-\frac{a^{2}}{4bD})\\ R_{+}=\frac{1}{2\pi }\sqrt{\left|\tilde{U^{\prime \prime }}(x_{+},y_{+},t)\tilde{U^{\prime \prime }}(x_{0},y_{0},t)\right|}\exp(\frac{\tilde{U}(x_{+},y_{+},t)-\tilde{U}(x_{0},y_{0},t)}{D})=\frac{a}{2\sqrt{2}\pi \gamma }\exp(-\frac{a^{2}}{4bD}) \end{array}$$

The transition of a particle from one potential well to another potential well and back again is defined as a cycle period, then the average escape rate of the particle transition back and forth can be obtained, that is, the Karmers escape rate is expressed as:(7)$$r_{k}=R_{-}+R_{+}=\frac{a}{\sqrt{2}\pi \gamma }\exp(-\frac{a^{2}}{4bD})$$

Therefore, when the signal include both periodic signal and noise excitation and the amplitude $A<A_{c}$, the bistable potential changes periodically [17].When the noise, the periodic signal, and the bistable system achieve synergy, the noise has positive effect on the signal, and some noise energy is transferred into the signal to increase the intensity of the periodic signal. When the average residence time $T_{k}=1/r_{k}$ of the Brownian particle in a potential well is equal to the change period of the potential function (i.e., half of the period $T=1/f_{m}$ of the excitation signal), the optimal SR will occur, and the *SNR* of the output will reach the maximum value [18].
(8)$$\frac{a}{\sqrt{2}\pi \gamma }\exp(-\frac{a^{2}}{4bD})=2f_{m}$$

Define a discriminant function $F(a,b,D,\gamma ,f_{m})$:(9)$$F(a,b,D,\gamma ,f_{m})=\frac{a}{2\sqrt{2}\pi \gamma f_{m}}\exp(-\frac{a^{2}}{4bD})$$

Obviously, to make the system produce optimal SR, this function should satisfy F=1.

For a sinusoidal signal with additive noise, the input *SNR* $SNR_{input}$ and output *SNR*$SNR_{output}$ of Equation (1) are:(10)$$\begin{array}{l} SNR_{input}=\frac{A^{2}}{4D}\\ SNR_{output}\approx \frac{aA^{2}\sqrt{2a}}{4bD^{2}\gamma }\exp(-\frac{a^{2}}{4bD}) \end{array}$$

The *SNR* gain is:(11)$$SNRI=\frac{SNR_{output}}{SNR_{input}}\approx \frac{a\sqrt{2a}}{bD\gamma }\exp(-\frac{a^{2}}{4bD})$$

The first order partial differential to the noise D is:(12)$$\frac{dSNRI}{dD}=\frac{\left(\frac{a^{3}\sqrt{2a}}{4b^{2}D}-\frac{a\sqrt{2a}}{b}\right)}{\gamma D^{2}}\exp(-\frac{a^{2}}{4bD})$$

Thus, when $D=a^{2}/(4b)$, the SNR gain reaches the maximum value, the optimal SR occurs, and the output *SNR* reaches the maximum value too [19].

### 2.2. Second-Order General Scale Transformation SR

The precondition of use of SR to enhance useful signal by noise is the input signal should satisfy the small parameter constraints, which means $A<A_{c}$, D<<1 and $f_{m}<<1$. However, most engineering signals are difficult to satisfy these conditions. Therefore, second-order general scale transformation (SGST) SR is proposed to deal with large parameter signal in engineering [20].

Introduce the scale coefficient m, then, τ=mt, z(τ)=x(t), substitute into Equation (1) and get:(13)$$\frac{d^{2}z(\tau )}{d\tau^{2}}+\gamma_{1}\frac{dz(\tau )}{d\tau }=a_{1}z(\tau )-b_{1}z^{3}(\tau )+A_{1}\cos(2\pi f_{m1}\tau )+\sqrt{2D_{1}m}\xi (\tau )$$

In which $a_{1}=\frac{a}{m^{2}}$, $b_{1}=\frac{b}{m^{2}}$, $\gamma_{1}=\frac{\gamma }{m}$, $f_{m1}=\frac{f_{m}}{m}$, $A_{1}=\frac{A}{m^{2}}$, $\sqrt{D_{1}}=\frac{\sqrt{D}}{m^{2}}$.

By choosing an appropriate value of m, the frequency $f_{m}$ of large parameter signal can be changed to its 1/m, thus, the high frequency signal can be converted into low frequency signal. Furthermore, the periodic signal and white Gaussian noise can be converted into $1/m^{2}$ of the original signal. Thus, the processed signal satisfies the small parameter requirement.

### 2.3. APSO Algorithm

For the same input signal, different system parameters will produce different SR effects. To achieve the optimal matching of weak fault signal, noise and SR system, PSO algorithm is used to adaptively obtain the optimal SR system parameters. In PSO algorithm, the optimized feasible solution can be abstracted as a particle in the m-dimensional search space, which only contains the position and velocity information. The update of particle velocity and position is as follows:(14)$$\left\{ \begin{array}{l} v_{ij}(t+1)=wv_{ij}(t)+c_{1}r_{1}(t)[pbest_{ij}(t)-x_{ij}(t)]+c_{2}r_{2}(t)[gbest_{ij}(t)-x_{ij}(t)]\\ x_{ij}(t+1)=x_{ij}(t)+v_{ij}(t+1)\begin{array}{cc} & 1\leq i\leq N,1\leq j\leq S \end{array} \end{array}\right.$$
where *N* represents the numbers of particles; *S* represents the dimension of the search space; $v_{ij}(t+1)$ represents the *j*-th dimension velocity of the *i*-th particle in the *t*-th iteration; $x_{ij}(t+1)$ represents the *j*-th dimension position of the *i*-th particle in the *t*-th iteration; *pbest* represents the best position of a single particle; *gbest* represents the best position of the particle group; $c_{1},\;c_{2}>0$ represents the learning factor; $r_{1}$ and $r_{2}$ are random numbers in the range of [0, 1]; *w* is the inertia weight factor [21]. To improve he global search and local optimization capabilities, the nonlinear dynamic inertia weight factor is expressed as:(15)$$w=\left\{ \begin{array}{ll} w_{\min}-\frac{\left(w_{\max}-w_{\min})(f-f_{\min}\right)}{f_{avg}-f_{\min}}, & f\leq f_{avg}\\ w_{\max}, & f>f_{avg} \end{array}\right.$$
where $w_{\max}$ and $w_{\min}$ are the maximum and minimum values of *w*, respectively; *f* represents the current objective function value of the particle; $f_{avg}$ and $f_{\min}$ represent the average target value and the minimum target value of the current particle swarm, respectively [22].

## 3. Second-Order Amplitude-Frequency Re-Scaling Match SR Based on CEI

### 3.1. Second-Order Amplitude Frequency Re-Scaling SR

In the general scale transform SR, the value of *m* not only affects the frequency of the useful signal, but also affects the amplitude and noise intensity of the useful signal. However, achieving the optimal SR requires the synergy of signal, noise and nonlinear system, only relying on a single parameter *m* cannot achieve good results. Therefore, to realize the optimal SR under the condition of large amplitude and large frequency, the amplitude transform coefficient ε and frequency transform coefficient *R* are introduced in this paper to realize the second-order amplitude frequency re-scaling SR. According to the Equation (1), its expression is derived as follows:(16)$$\frac{d^{2}x(t^{\prime })}{d{t^{\prime }}^{2}}+\gamma \frac{dx(t^{\prime })}{dt^{\prime }}-ax+bx^{3}=\varepsilon A\cos(2\pi \frac{f_{m}}{R}t^{\prime })+\sqrt{2\varepsilon^{2}D}\xi (t^{\prime })$$
where $t^{\prime }=Rt$ is the transformed time scale, $x(t^{\prime })$ is the system output represented by the scale $t^{\prime }$, the transformed frequency is 1/R, and ε is the amplitude transform coefficient used to linearly amplify or reduce the useful signal.

### 3.2. Parameter Matching Principle

Equations (9) and (12) shows that the optimal condition of the second-order amplitude frequency re-scaling SR is as following:(17)$$\left\{ \begin{array}{l} F(a,b,D,f_{m},\gamma ,\varepsilon ,R)=\frac{aR}{2\sqrt{2}\pi \gamma f_{m}}\exp(-\frac{a^{2}}{4b\varepsilon^{2}D})=1\\ D=a^{2}/(4b) \end{array}\right.$$

The optimal matching relationship between signal frequency, noise intensity and system parameters is obtained as:(18)$$\left\{ \begin{array}{l} a=\frac{2\sqrt{2}\pi \gamma f_{m}e}{R}\\ b=\frac{a^{4}}{4\varepsilon^{2}D} \end{array}\right.$$
where e is Euler number.

In the traditional second-order amplitude-frequency-rescaling SR, it is necessary to optimize the system parameters *a* and *b*, the damping factor γ, the amplitude transformation coefficient ε and the frequency transformation coefficient *R* at the same time, and the parameter adjustment range is usually determined by experience, which would increase the computational complexity of the algorithm, and decrease the accuracy of the optimal parameters. Through the optimal parameter matching principle, the system parameters *a* and *b* can be transformed into expressions related to γ and ε. According to the optimal values of γ and ε, *a* and *b* can be determined, which can simplify the optimization algorithm.

(1)determination of the range of *R*

The selection of *R* is related to the calculation step *h* of the SR. Set the sampling frequency of the input signal is $f_{s}$, and the frequency transform coefficient is *R*. Therefore, the compressed sampling frequency is $f_{sr}=f_{s}/R$, and the SR calculation step size is $h=1/f_{sr}=R/f_{s}$.If *R* is too small, the SR cannot satisfy the adiabatic approximation theory, and is difficult to occur resonance; if *R* is too large, the calculation step *h* is too large, which leads to the divergence of the system response, and would be difficult to extract the feature information.

A simulated signal is used to express the influence of R to the output *SNR*. The simulated signal is periodic signal with noise, and sampling frequency is $f_{s}=10$ kHz, amplitude is A=1, noise intensity is D=0.5, characteristic frequency is $f_{m}=100$ Hz. Set γ=0.3 and ε=0.15. The output *SNR* of the proposed method with *R* is shown in Figure 2.

Figure 2 shows that with the increase in *R*, the output *SNR* increases firstly and then tends to be stable. When R<2500, the output *SNR* increases greatly with *R*; when R>3500, the output *SNR* tends to be stable and reduces slowly. It can be seen from the description of Equation (18) that for the optimal parameter a, the change in the value of R will affect the change in the value of the damping factor γ. The value of γ is optimized through the APSO algorithm, so the value of R will only affect the optimization range of R, and will not affect the optimal parameters of SR. Therefore, R can take a value in range of 2500~3500.

(2)determination of the range of ε

The amplitude of input signal is transformed with the adjustment of ε, and there is an adjustable range $\varepsilon \in (\varepsilon_{\min},\varepsilon_{\max})$ to make the input signal satisfies $\varepsilon A<A_{c}$. As the amplitude *A* in the engineering signal is relatively large, so the range of ε is set to 0<ε<0.5.

(3)determination of the range of γ

The damping factor γ is restricted in the interval $\left[0,2\sqrt{2a}\right]$ according to Lyapunov’s stability analysis, which provides the constraints for parameter tuning [23]. Therefore, the optimization range of ε and γ is:(19)$$\left\{ \begin{array}{l} 0<\gamma <\frac{16\sqrt{2}\pi f_{m}e}{R}\\ 0<\varepsilon <0.5 \end{array}\right.$$

### 3.3. CEI Based on BP Neural Network

#### 3.3.1. Single Index Analysis

In the theoretical analysis of SR, one or more parameters need to be adjusted to obtain the optimal effect. Therefore, many adaptive SR methods have been proposed, in which an adaptive optimization index must be built to evaluate the effect of the SR system. Usually, the output signal *SNR* or *SNR* gain is used as the evaluation index. *SNR* is defined as follows:(20)$$SNR=10\log_{10}\frac{P_{f_{m}}}{P_{i}-P_{f_{m}}}$$
where *N* is the length of signal, $P_{f_{m}}$ represents the power of the useful signal, and $P_{i}$ represents the power of the signal. Therefore, the higher *SNR* is, the better the SR denoising effect is.

However, the calculation of *SNR* needs the information useful signal, which is usually unknown in engineering applications. Therefore, an adaptive index which can evaluate the effect of SR system and has the similar performance to *SNR* is needed to be built. A new comprehensive quantitative index (SQI) to evaluate the effect of SR was proposed in Reference [24]. The SQI index is obtained by merging the six indexes of *PSK*, *CC*, *PSNR*, *SSIM*, *RMSE* and *SMO* through the BP neural network. However, the equations for calculating *PSNR* and *RMSE* are similar, which results in similar effects for *PSNR* and *RMSE*. When the *SNR* of the output signal is high, the *SMO* index is not sensitive to *SNR*. Therefore, the approximate entropy is introduced to replace the *PSNR* index and the *SMO* index. The approximate entropy of a periodic signal is not affected by amplitude and phase but is only related to frequency and *SNR*. In this paper, the power spectrum kurtosis, correlation coefficient, structural similarity, root mean square error and approximate entropy are introduced to be fused:(1)Power Spectrum Kurtosis (*PSK*)

*PSK* reflects the sharpness of the power spectrum of the output signal of SR [25]. Assuming $x(i)=[x_{1},x_{2},\cdots ,x_{N}]$ is the SR system output (with *N* data points) and $P=[P_{1},P_{2},\cdots ,P_{M}]$ is the power spectrum of *x*(*i*) obtained by M-point discrete Fourier transform, the *PSK* is defined as:(21)$$PSK=\frac{\frac{1}{M/2}\sum_{i=1}^{M/2}{\left(P_{i}-\bar{P}\right)}^{4}}{{\left(\frac{1}{M/2}\sum_{i=1}^{M/2}{\left(P_{i}-\bar{P}\right)}^{2}\right)}^{2}}$$
where $\bar{P}$ is the mean of *P*.

The larger the value of *PSK* is, the better the filtering effect is. Moreover, *PSK* is positively related to the *SNR*.

(2)Correlation Coefficient (*CC*)

*CC* reflects the correlation between the output signal of SR and the original input signal [25]. The *CC* is defined as:(22)$$CC=\frac{\sum_{i=1}^{N}\left(x(i)-\bar{x})(s(i)-\bar{s}\right)}{\sqrt{\sum_{i=1}^{N}{\left(x(i)-\bar{x}\right)}^{2}\sum_{i=1}^{N}{\left(s(i)-\bar{s}\right)}^{2}}}$$
where s(i) and x(i) are the discrete forms of the original input signal *s*(*t*) and output signal *x*(*t*), respectively, $\bar{s}$ and $\bar{x}$ are the mean values of s(i) and x(i), respectively.

The larger the value of *CC* is, the greater the correlation is. And *CC* is positively related to the *SNR*.

(3)Structural Similarity (*SSIM*)

*SSIM* reflects the similarity between the output signal of SR and the original input signal. The *SSIM* is defined as:(23)$$SSIM=\frac{\left(2\bar{x}\times \bar{s}+1)(2\sigma_{xs}+1\right)}{\left({\bar{x}}^{2}+{\bar{s}}^{2}+1)(\sigma_{x}^{2}+\sigma_{s}^{2}+1\right)}$$
where $\sigma_{s}^{2}$ and $\sigma_{x}^{2}$ are the variances of s(i) and x(i), respectively; $\sigma_{xs}$ is the covariance of s(i) and x(i).

The larger the value of SSIM is, the greater the similarity is. *SSIM* is positively related to the *SNR*.

(4)Root Mean Square Error (*RMSE*)

*RMSE* reflects the degree of dispersion between the output signal of SR and the original input signal. The *RMSE* is defined as:(24)$$RMSE=\sqrt{\frac{1}{N}\sum_{i=1}^{N}{\left(s(i)-x(i)\right)}^{2}}$$

The smaller the value of *RMSE* is, the smaller the discrete degree is. *RMSE* is inversely related to the *SNR*.

(5)Approximate Entropy (*ApEn*)

*ApEn* reflects the similarity between the output signal of SR and the original input signal [26]. The flowchart of calculating *ApEn* is shown in Figure 3.
Figure 3The flowchart of calculating *ApEn*.
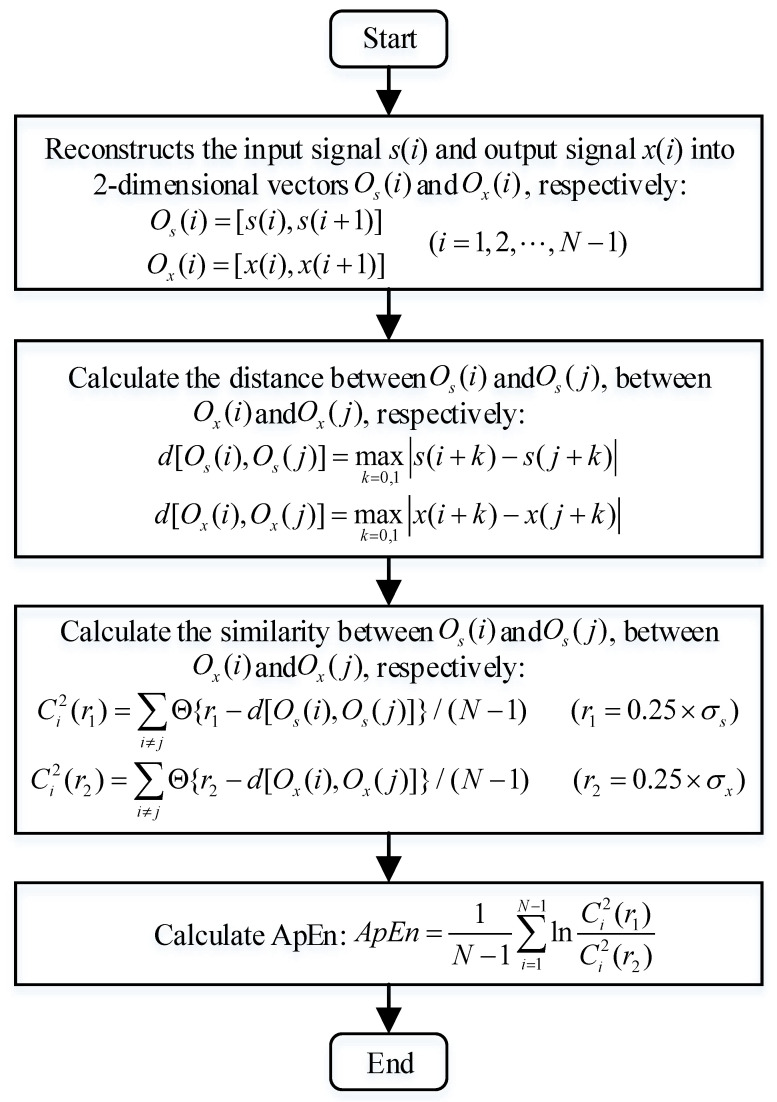

where $\sigma_{s}$ is the standard deviation of *s*(*i*), $\sigma_{x}$ is the standard deviation of *x*(*i*).

The smaller *ApEn* is, the greater the similarity is. Furthermore, it is inversely related to the *SNR*.

To further verify the above five indicators as the evaluation indexes of SR, the following simulation is introduced. The simulation signal is:(25)$$s(t)=A\cos(2\pi f_{m}t)+\sqrt{2D}\xi (t)$$
where the sampling frequency $f_{s}=2000$ Hz, the amplitude A=1, the characteristic frequency $f_{m}=20$ Hz, the signal length N=2000, the noise intensity D∈(0,10), the interval is 0.05, and there are 200 groups of different noise intensities. The trends of the six indicators under different noise intensities are shown in Figure 4.The index values under different noise intensities (three groups before and after the data are selected) are shown in Table 1.

Figure 4 shows that *SNR*, *PSK*, *CC* and *SSIM* decrease with the increase in noise intensity, while *RMSE* and *ApEn* increase with the increase in noise intensity. The specific values in Table 1 can also reflect these trends. The results further verify the relationship between the five indicators and *SNR*. So, the above five indicators can be used as the evaluation indicators of SR.

#### 3.3.2. Index Fusion Based on BP Neural Network

Dueto the serious noise interference in the actual engineering signal, the phase of the output signal of the SR system would be changed, and the output of the SR system may resonate at different frequencies. A single index cannot obtain good performance. Therefore, a new CEI based on BP neural network is proposed, which is consisted of *PSK*, *CC*, *SSIM*, *RMSE* and *ApEn*. The flow of CEI based on BP neural network is shown in Figure 5.

The steps of CEI based on BP neural network are summarized as follows:

Step 1: Parameter normalization. The five-column data of PSK, CC, SSIM, RMSE and ApEn in Table 1 form a 5-dimensional matrix:(26)$${\left[\begin{array}{ccccc} 913.5260 & 0.9151 & 0.9954 & 0.3182 & 0.5156\\ 827.3234 & 0.8404 & 0.9924 & 0.4569 & 0.7350\\ 756.7657 & 0.7851 & 0.9843 & 0.5524 & 0.8950\\ \vdots & \vdots & \vdots & \vdots & \vdots \\ 4.7536 & 0.1235 & 0.0059 & 4.4385 & 1.8435\\ 4.6039 & 0.1219 & 0.0057 & 4.4473 & 1.8467\\ 4.3926 & 0.1215 & 0.0056 & 4.4978 & 1.8483 \end{array}\right]}_{\left(200\times 5\right)}$$

Equation (26) is normalized to obtain the input matrix X:(27)$$X={\left[\begin{array}{ccccc} 1 & 1 & 1 & 0 & 0\\ 0.9052 & 0.9059 & 0.9530 & 0.0332 & 0.1646\\ 0.8276 & 0.8363 & 0.9083 & 0.0560 & 0.2847\\ \vdots & \vdots & \vdots & \vdots & \vdots \\ 0.0004 & 0.0025 & 0.0003 & 0.9858 & 0.9964\\ 0.0002 & 0.0004 & 0.0001 & 0.9879 & 0.9988\\ 0 & 0 & 0 & 1 & 1 \end{array}\right]}_{\left(200\times 5\right)}$$
where X=[PSK,CC,SSIM,RMSE,ApEn].

The *SNR* data in Table 1 is formed into a 1-dimensional matrix to obtain:(28)$${\left[\begin{array}{ccccccc} 22.9475 & 16.9511 & 13.4987 & \cdots & -25.6313 & -27.1864 & -28.1692 \end{array}\right]}_{\left(1\times 200\right)}$$

Equation (28) is reverse normalized to obtain the output matrix Y:(29)$$Y={\left[\begin{array}{ccccccc} 0 & 0.0193 & 0.0496 & \cdots & 0.8152 & 0.8827 & 1 \end{array}\right]}_{\left(1\times 200\right)}$$
where Y=CEI.

Step 2: Initialize the BP neural network. The topology diagram of BP neural network is shown in Figure 6.

There are n=5 nodes in the input layer of the BP neural network, which are PSK, CC, SSIM, RMSE and ApEn. The hidden layer has l=4 nodes, and the final output layer has only m=1 node, which is CEI. $w_{ij}$, $w_{ik}$, a and b are initialized, where $w_{ij}$ is the connection weight between the input layer and the hidden layer, $w_{ik}$ is the connection weight between the hidden layer and the output layer, a is the hidden layer threshold, and b is the output layer threshold.

Step 3: Calculate the hidden layer output. The hidden layer output H can be calculated according to Equation (30):(30)$$H_{j}=f(\sum_{i=1}^{n}w_{ij}x_{i}-a_{j})\begin{array}{cc} & \left(j=1,2,\cdots ,l\right) \end{array}$$
where f is the activation function of hidden layer neurons. The Sigmoid activation function is used in this paper, and the expression is shown in Equation (31):(31)$$f(x)=\frac{1}{1+e^{-x}}$$

Step 4: Calculate the output layer output. The predicted output O of the BP neural network is calculated by Equation (32):(32)$$O_{k}=\sum_{j=1}^{l}H_{j}w_{jk}-b_{k}\begin{array}{cc} & k=1,2,\cdots ,m \end{array}$$

Step 5: Calculate error. The prediction error e is calculated by Equation (33):(33)$$e_{k}=Y_{k}-O_{k}\begin{array}{cc} & k=1,2,\cdots ,m \end{array}$$

Step 6: Update weights and thresholds. The network weights $w_{ij}$, $w_{ik}$ and neuron thresholds a, b are updated according to Equation (34):(34)$$\left\{ \begin{array}{l} w_{ij}=w_{ij}+\eta H_{j}(1-H_{j})x(i)\sum_{k=1}^{m}w_{jk}e_{k}\\ w_{jk}=w_{jk}+\eta H_{j}e_{k}\\ a_{j}=a_{j}+\eta H_{j}(1-H_{j})\sum_{k=1}^{m}w_{jk}e_{k}\\ b_{k}=b_{k}+e_{k}\begin{array}{ccc} & & i=1,2,\cdots ,n;\;j=1,2,\cdots ,l;\;k=1,2,\cdots ,m \end{array} \end{array}\right.$$
where η=0.01 is the learning rate.

Step 7: Judging the end of the algorithm iteration. When the prediction error reaches the error precision requirement or the calculation number of algorithm reaches the set number, where the error precision is $e=10^{-4}$ and the maximum iteration number of algorithm is t=100, finish the training. Otherwise, go back to Step 3.

Step 8: Output prediction matrix. After the BP neural network algorithm is trained, the prediction output CEI matrix can be obtained.

Simulation is introduced to verify the accuracy of the BP neural network model. Firstly, normalize the 200 sets of data in Table 1, and randomly select 180 sets as training sets, the remaining 20 sets as the test sets. Secondly, construct input matrix *X* by *PSK*, *CC*, *SSIM*, *RMSE* and *ApEn*, output matrix Y by *SNR*. Thirdly, train the model through training sets. Set the iteration numbers of BP neural network to 100 times, and the error accuracy to $10^{-4}$. The results of CEI based on BP neural network are shown in Figure 7.

Figure 7a shows that the training is completed after 69 iterations, and the accuracy meets the set accuracy expectations. Figure 7b shows that the error value is in the range of −0.008~0.01, which is very small. Figure 7c shows that the target output curve is basically consistent with the predicted output curve, which means CEI is reliable.

#### 3.3.3. Performance Evaluation of CEI

Proposed CEI is used as a signal quality evaluation index, the smaller the value, the less the noise interference component in the signal, which means the better the filtering effect of SR. To verify the applicability of CEI as an evaluation index, a comparative experiment is carried out with the *SNR* in the SR. The expression of the simulated signal is shown in Equation (25). Set the sampling frequency $f_{s}=2000$ Hz, the amplitude A=1, the characteristic frequency $f_{m}=20$ Hz, the noise intensity D∈(0,5); and set the SR system parameter R=2500, ε=0.10, γ=0.15, the comparison between *SNR* and CEI of the SR system is shown in Figure 8.

Figure 8 shows that when *SNR* is used as the evaluation index, with the increase in the noise intensity D, *SNR* increases until the optimal resonance point, then decreases; when CEI is used as the evaluation index, with the increase in the intensity D, CEI decreased until the optimal resonance point, then increased. The optimal resonance points are all near the noise intensity D=1.2, and the trend of CEI is almost opposite to *SNR*, So, CEI can be used to determine the effect of SR. The smaller the value of CEI, the better the filtering effect of SR.

### 3.4. SAFRM Adaptive SR Based on CEI

The SAFRM adaptive SR based on CEI is proposed to optimize the parameters of the SR system. Set the target range of the parameters need to be optimized, search the optimal parameters at the minimum CEI through APSO method, detect the fault signal with the optimal SR system. The flowchart of the SAFRM adaptive SR based on CEI is shown in Figure 9.

The main steps of the process are as follows:

Step 1: Signal preprocessing. The signal is preprocessed by band-pass filtering or envelope extraction, which shows the periodicity of the signal;

Step 2: Drive frequency estimation and noise variance estimation. In this paper, the value of the driving frequency $f_{m}$ is set to 100; the noise intensity *D* is estimated according to the principle of maximum likelihood estimation (MLE) [27];

Step 3: Initialize parameters and target function. According to the restriction of Equation (19), initialize the search range of γ and ε;

Step 4: Optimization method. Search the optimal parameter group $\left(\gamma_{opt},\varepsilon_{opt}\right)$ at the minimum CEI through APSO method:(35)$$\left(\gamma_{opt},\varepsilon_{opt})=\arg\min(CEI(\gamma ,\varepsilon )\right)$$

Step 5: Construct the SR system based on the optimal parameter group. Substituting the optimal parameter group in Step 4 into Equation (18), calculating the system parameters *a* and *b*, and obtaining the optimal detection result of the original input signal. Through the proposed method, the fault characteristic frequency can be extracted.

## 4. Simulation

### 4.1. Performance Comparison of CEI and SNR

To verify the filtering performance of the SR based on proposed CEI, the simulation is compared with the SR based on *SNR*. The SAFRM adaptive SR method proposed in this paper and the SGST adaptive SR method are used to compare. The input signal is expressed by Equation (25). Set the sampling frequency $f_{s}=2000$ Hz, the amplitude A=1, the characteristic frequency $f_{m}=20$ Hz, the noise intensity D=2; the SR frequency transformation coefficient R=2500. The comparison between CEI and *SNR* is shown in Figure 10.

According to the comparison of Figure 10c–f, it can be seen that the filtering performance of the SAFRM adaptive SR proposed in this paper is better than the traditional SGST adaptive SR. According to the comparison of Figure 10c–f, the CEI value of the SR output signal with CEI as the evaluation index is slightly higher than the CEI value of the SR output signal with *SNR* as the evaluation index. However, the CEI value of the SR output signal with CEI as the evaluation index is much lower than the CEI value of the noise signal. These all show that CEI can be used as the evaluation index.

### 4.2. Performance Comparison of Two SR Methods

To verify the effectiveness of the proposed method, the simulation is introduced with simulated outer ring fault signals. The bearing outer ring fault simulation formula is as follows:(36)$$\left\{ \begin{array}{l} s(t)=s_{0}(t)+n(t)\\ s_{0}(t)=A\cos(2\pi f_{n}t)\times \exp(-B{\left(t-i(t)/f_{m}\right)}^{2})\\ n(t)=\sqrt{2D}\xi (t)\\ i(t)=floor[t\times f_{m}] \end{array}\right.$$
where $s_{0}(t)$ is the outer ring fault vibration simulation signal; ξ(t) is Gaussian white noise with zero-mean and unit-variance; *A* is the amplitude; *B* is the attenuation coefficient; *D* is the noise intensity; $f_{n}$ is the natural frequency of the bearing; $f_{m}$ is the fault characteristic frequency of the outer ring of the bearing; *i*(*t*) is the number of repetitions; *floor*() is the function of rounding down.

In the simulation signal, the sampling frequency $f_{s}=10$ kHz, the number of sampling points N=5000; Set:(37)$$A=0.5,\;B=10f_{s},\;f_{n}=2000\mathrm{Hz},\;f_{m}=100\mathrm{Hz},\;D=0.5$$

The simulated signals are shown in Figure 11.

Figure 11a,b show that the characteristic frequency $f_{m}$ and natural frequency $f_{n}$ can be clearly extracted with no noise. Figure 11c,d show that only the natural frequency $f_{n}$ can be extracted, the characteristic frequency $f_{m}$ is submerged in the noise.

To extract the characteristic frequency, different methods are used to compare. The parameters of the APSO algorithm are set as: N=50, t=100. In SGST adaptive SR, the optimization range of parameters is set to a∈(0,2), b∈(0,2), γ∈(0,1), and the search space dimension is S=3. In SAFRM adaptive SR, the optimization range of the parameters is calculated according to Equation (19): γ∈(0,6.44), ε∈(0,0.5), and the search space dimension is S=2. The results of with different methods are shown in Figure 12.

The Hilbert envelope signal and envelope spectrum obtained by the Hilbert transform demodulation of noisy signal are shown in Figure 12a,b. In the envelope spectrum, the fault characteristic frequency $f_{m}$ is not obvious. The output signal and spectrum of SGST adaptive SR are shown in Figure 12c,d. According to optimal parameters, the output signal is improved, and the fault characteristic frequency $f_{m}$ and its double frequency is enhanced, but the noise component is still obvious. The output signal and spectrum obtained by the proposed method are shown in Figure 12e,f. According to optimal parameters, the output signal is improved clearly, and the characteristic frequency $f_{m}$ can be clearly extracted from the spectrum, and most of the noise energy is converted into useful signal energy.

The results of the three methods are shown in Table 2.

Table 2 shows that the amplitude at the characteristic frequency $f_{m}$ in the spectrum of the noisy signal is 0.0164V. Through the proposed method, the amplitude at the characteristic frequency $f_{m}$ in the output spectrum is increased to 2.2560 V, with an increase of 137.56 times, which is better than the Hilbert envelope demodulation method and the SGST adaptive SR method. The CEI of the noisy signal is 0.6243, and the CEI of the output signal obtained by proposed reduced to 0.0852, with a decrease of 0.5391, which is better than the Hilbert envelope demodulation method and the SGST adaptive SR method. Therefore, it can be concluded that the SR system proposed in this paper can effectively extract the features of weak fault signal and convert most of the noise energy into characteristic signal energy; moreover, the filtering effect is better than the SGST adaptive SR method.

## 5. Application

The bearing experimental data is from Case Western Reserve University (CWRU). In the experiment, the type of the drive end bearing is 6205-2RS JEM SKF, the outer ring fault and inner ring fault data are selected for the experiment, the sampling frequency is 12 kHz, and the parameters of the faulty bearing are shown in Table 3.

The bearing outer ring fault signals are shown in Figure 13.

The original signal and spectrum are shown in Figure 13a,b. They show that the shock component caused by the fault is obvious, and the noise interference is small, which cannot reflect the waveform characteristics of the early bearing fault. To simulate the bearing early weak fault signal, white Gaussian noise with noise intensity D=2 is added to the original signal. The noisy signal and spectrum are shown in Figure 13c,d. The fault impact component is not obvious in noisy signal, and the characteristic frequency of the outer ring fault is completely submerged by noise in the spectrum.

Using different methods to extract the fault signal, the parameters of the APSO algorithm are set as: N=50, t=100. In SGST adaptive SR, the optimization range of parameters is set to a∈(0,2), b∈(0,2), γ∈(0,1), and the search space dimension is S=3. In SAFRM adaptive SR, the optimization range of the parameters is calculated according to Equation (19): γ∈(0,6.44), ε∈(0,0.5), and the search space dimension is S=2. The results of the different methods are shown in Figure 14.

The Hilbert envelope signal and envelope spectrum of noisy signal are shown in Figure 14a,b. In the envelope spectrum, the outer ring fault characteristic frequency and its frequency doubling components can be observed, but the amplitude is lower. The output signal and spectrum of SGST adaptive SR are shown in Figure 14c,d. The characteristic frequency is enhanced, and the amplitudes at the double frequency and triple frequency are also more obvious, but there are still a lot of noise components. The output signal and spectrum obtained by the proposed method are shown in Figure 14e,f. The characteristic frequency can be clearly seen from the spectrum, and the noise component is very small, which shows that most of the noise energy is converted into useful signal energy.

The bearing inner ring fault signals are shown in Figure 15.

The original signal and spectrum are shown in Figure 15a,b. They show that the shock component caused by the fault is obvious, and the noise interference is small, which cannot reflect the waveform characteristics of the early bearing fault. To simulate the bearing early weak fault signal, white Gaussian noise with noise intensity D=1 is added to the original signal. The noisy signal and spectrum are shown in Figure 15c,d. The fault impact component is not obvious in noisy signal, and the characteristic frequency of the inner ring fault is completely submerged by noise in the spectrum.

Using different methods to extract the fault signal, the parameters of the APSO algorithm are set as: N=50, t=100. In SGST adaptive SR, the optimization range of parameters is set to a∈(0,2), b∈(0,2), γ∈(0,1), and the search space dimension is S=3. In SAFRM adaptive SR, the optimization range of the parameters is calculated according to Equation (19): γ∈(0,5.52), ε∈(0,0.5), and the search space dimension is S=2. The results of different methods are shown in Figure 16.

The Hilbert envelope signal and envelope spectrum of noisy signal are shown in Figure 16a,b. In the envelope spectrum, the inner ring fault characteristic frequency and its frequency doubling components can be observed, but the amplitude is lower. The output signal and spectrum of SGST adaptive SR are shown in Figure 16c,d. The characteristic frequency is enhanced, and the amplitude at the double frequency is also more obvious, but there are still a lot of noise components. The output signal and spectrum obtained by the proposed method are shown in Figure 16e,f. The characteristic frequency can be clearly seen from the spectrum, and the noise component is very small, which shows that most of the noise energy is converted into useful signal energy.

The results of the three methods are shown in Table 4.

Table 4 shows that the bearing early weak fault signal is interfered by strong noise, the amplitude at the characteristic frequency is very small, and the fault features are completely submerged by the noise. In the feature extraction of the bearing early weak fault signal, the weak fault feature cannot be effectively enhanced by the traditional Hilbert envelope demodulation method, and the noise in the spectrum still dominates. Using the SGST adaptive SR method, the bearing weak fault characteristic signal can be enhanced, and the fault characteristics can be extracted, but the amplitude at the characteristic frequency is not higher than that of some noise components. The output signal waveform obtained by the proposed method is smoother, and the amplitude at the characteristic frequency in the spectrum increases significantly, which is significantly higher than the noise component. It shows that most of the noise energy is converted into useful signal energy through the SR system proposed in this paper, and the bearing early weak fault characteristics are effectively enhanced. Comparing the values of CEI, it can be seen that the CEI of the output signal obtained by the proposed method is the smallest, which further verifies that the filtering effect of the proposed method is better than the other two methods.

## 6. Conclusions

Aiming at the problem of early weak fault feature extraction of bearings, this paper proposes a SAFRM adaptive SR method based on CEI, which can improve the ability to utilize and transform noise energy. Simulations verify the effectiveness of the proposed method in weak feature extraction, and applications also verify the important application value. The specific conclusions are as follows:

Aiming at the difficulty of single scale transform coefficient to match the signal amplitude and characteristic frequency at the same time, a second-order amplitude-frequency re-scaling match (SAFRM) SR method is proposed, which introduces the amplitude transform coefficient and frequency transform coefficient to realize the optimal match of signal, noise and nonlinear system. Aiming at the difficult of the *SNR* calculation in engineering signal, a new comprehensive evaluation index (CEI) is proposed, which uses the BP neural network to fuse five indexes of power spectrum kurtosis, correlation coefficient, structural similarity, root mean square error and approximate entropy. This CEI can overcome the reliance on unknown characteristic frequency, and the SR system can obtain the optimal parameters when CEI obtains the minimum value. So, through the CEI-based adaptive weight particle swarm optimization (APSO) algorithm, the optimal parameter values of SR system can be obtained; thus, through this optimal SR system, weak fault characteristic signal can be extracted.

(1)Aiming at the difficulty of single scale transform coefficient to match the signal amplitude and characteristic frequency at the same time, a second-order amplitude-frequency re-scaling match SR method is proposed, which introduces the amplitude transform coefficient and frequency transform coefficient to realize the optimal match of signal, noise and nonlinear system.(2)Aiming at the difficult of the *SNR* calculation in engineering signal, a new comprehensive evaluation index is proposed, which uses the BP neural network to fuse five indexes of power spectrum kurtosis, correlation coefficient, structural similarity, root mean square error and approximate entropy. This CEI can overcome the reliance on unknown characteristic frequency, and the SR system can obtain the optimal parameters at minimum CEI. Through the optimal SR system based on the proposed method, a weak fault characteristic signal can be extracted.

## Figures and Tables

**Figure 1 sensors-22-06644-f001:**
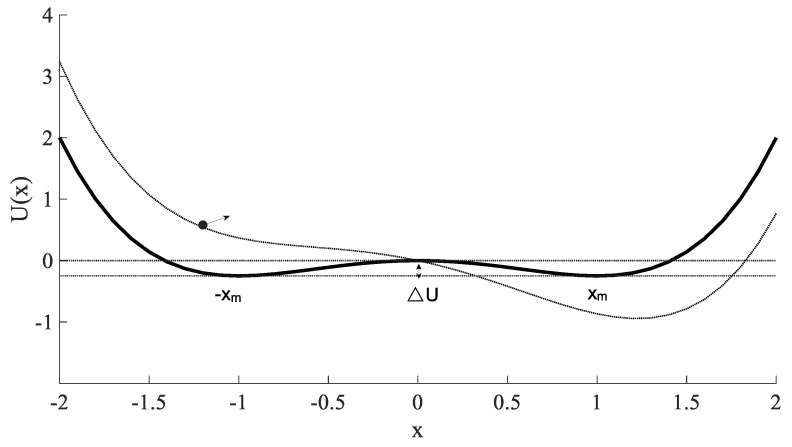
Bistable potential function U(x) (a=b=1 ).

**Figure 2 sensors-22-06644-f002:**
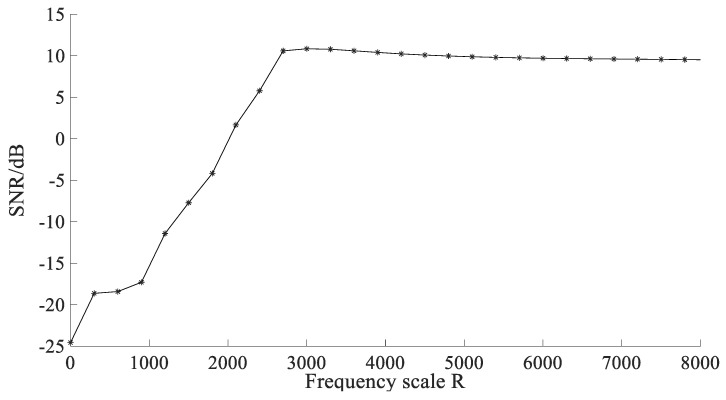
The output *SNR* of the proposed method with *R*.

**Figure 4 sensors-22-06644-f004:**
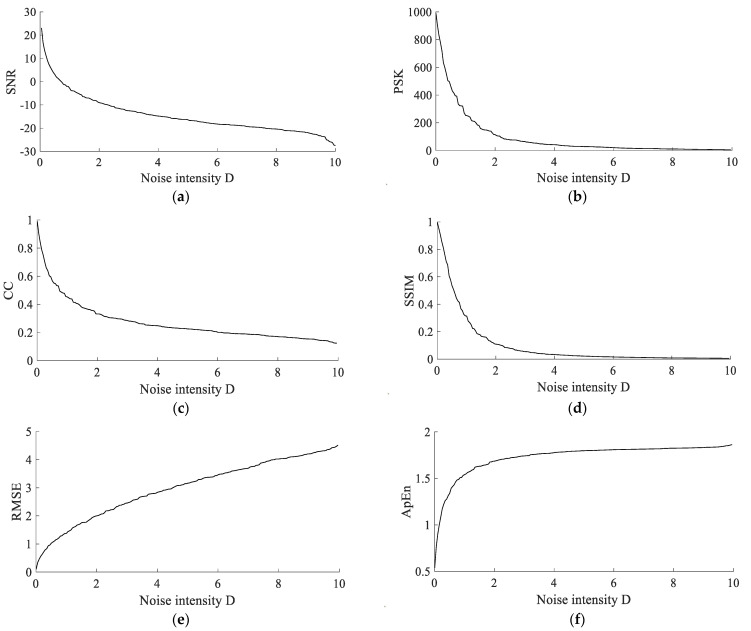
Trends of the six indicators under different noise intensities. (**a**) Trends of *SNR*. (**b**) Trends of *PSK*. (**c**) Trends of *CC*. (**d**) Trends of *SSIM*. (**e**) Trends of *RMSE*. (**f**) Trends of *ApEn*.

**Figure 5 sensors-22-06644-f005:**
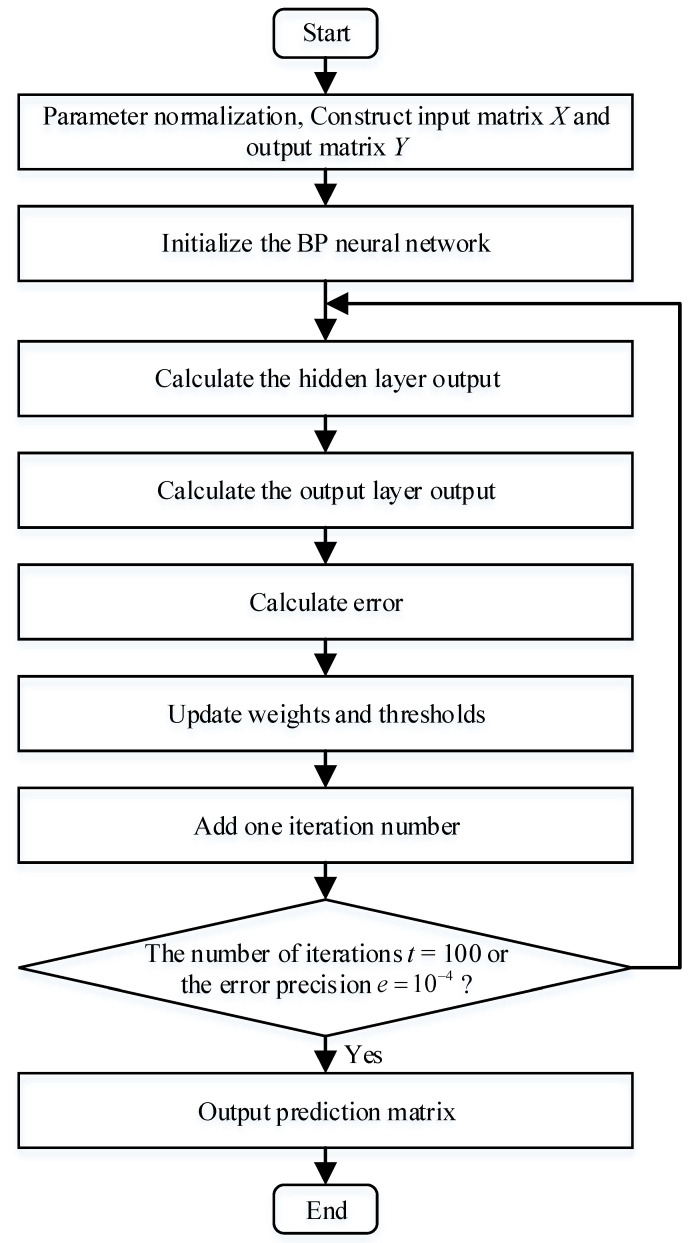
The flow of CEI based on BP neural network.

**Figure 6 sensors-22-06644-f006:**
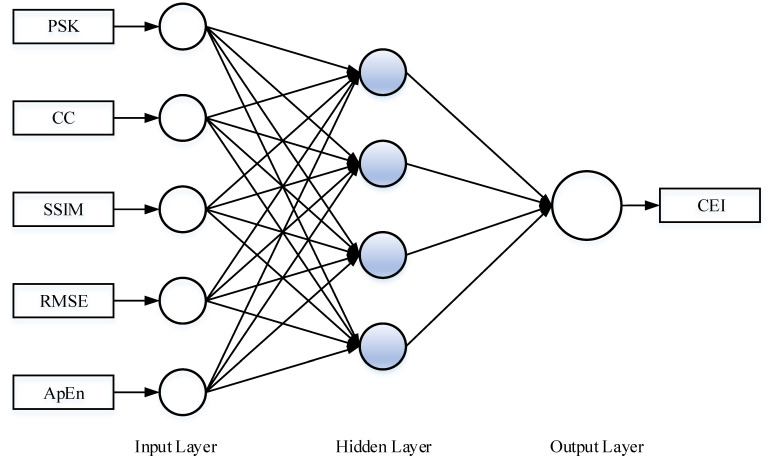
The topology diagram of BP neural network.

**Figure 7 sensors-22-06644-f007:**
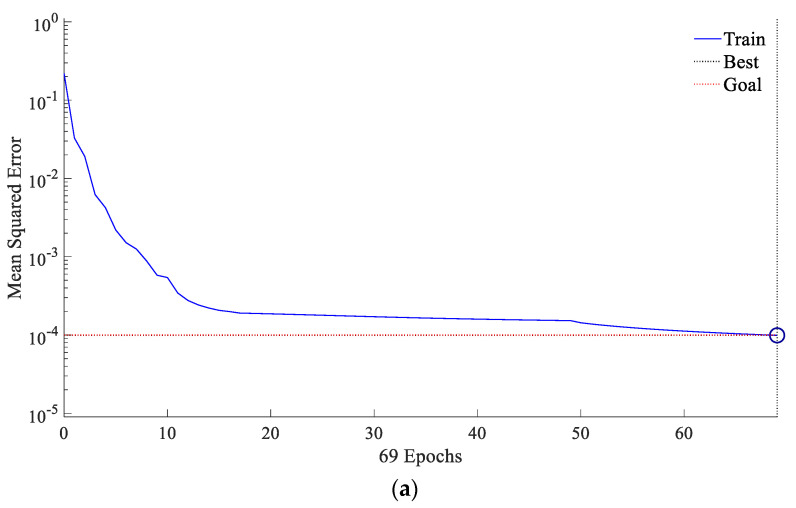
The analysis diagram of the accuracy of CEI constructed by BP neural network: (**a**) the iterative accuracy curve of the BP neural network; (**b**) the prediction error of the BP neural network; (**c**) the comparison result between the predicted output of the BP neural network and the target output.

**Figure 8 sensors-22-06644-f008:**
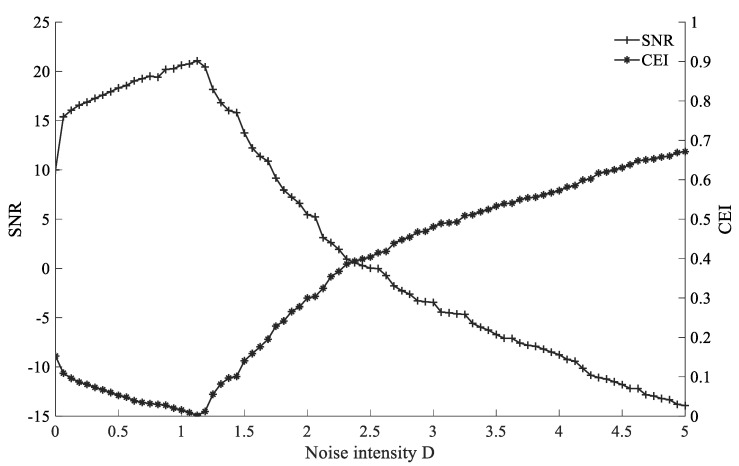
Comparison between *SNR* and CEI of the SR system.

**Figure 9 sensors-22-06644-f009:**
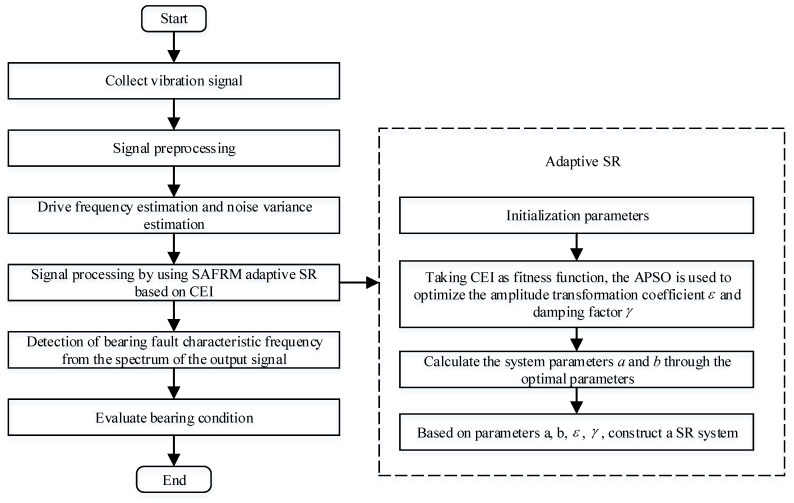
The flowchart of the SAFRM adaptive SR based on CEI.

**Figure 10 sensors-22-06644-f010:**
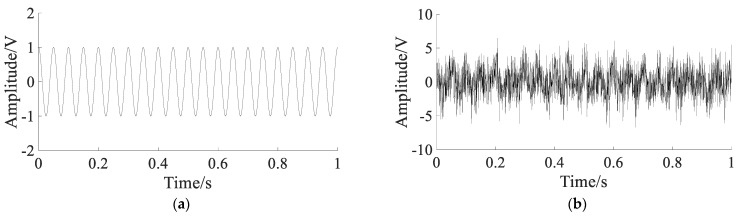
The comparison between CEI and *SNR*: (**a**) pure signal: CEI=0.0020; (**b**) noise signal: CEI=0.4812; (**c**) output signal of SGST adaptive SR based on *SNR*: a=0.0248, b=0.0067, γ=2.3749, CEI=0.1021; (**d**) output signal of SGST adaptive SR based on CEI: a=0.0168, b=0.0382, γ=2.1070, CEI=0.1187; (**e**) output signal of SAFRM adaptive SR based on *SNR*: γ=0.05, ε=0.13, CEI=0.0568; (**f**) output signal of SAFRM adaptive SR based on CEI: γ=0.13, ε=0.16, CEI=0.0689.

**Figure 11 sensors-22-06644-f011:**
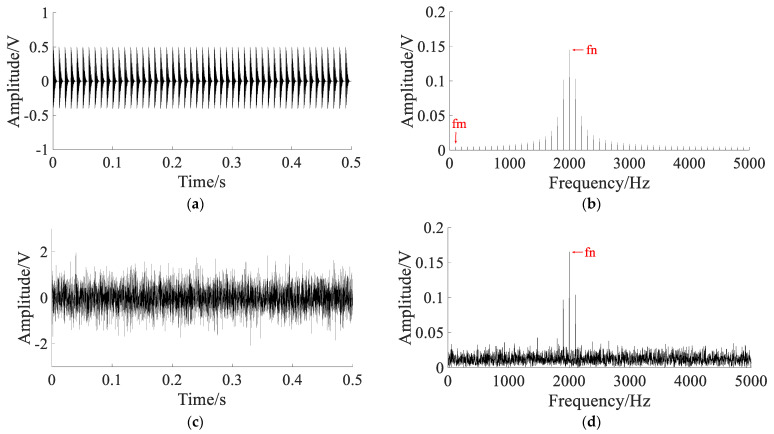
The simulated signals: (**a**) pure signal; (**b**) spectrum of pure signal; (**c**) noisy signal; (**d**) spectrum of noisy signal.

**Figure 12 sensors-22-06644-f012:**
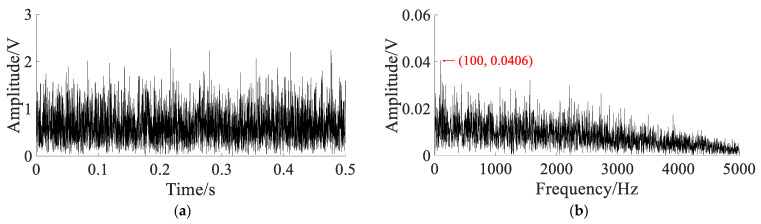
The results of with different methods: (**a**) Hilbert transform envelope demodulation signal; (**b**) Hilbert envelope spectrum; (**c**) output signal of SGST adaptive SR: a=0.4547, b=0.0012, m=3000, γ=0.3996; (**d**) output signal spectrum of SGST adaptive SR; (**e**) output signal of SAFRM adaptive SR: R=3000,γ=0.0112, ε=0.2663; (**f**) output signal spectrum of SAFRM adaptive SR.

**Figure 13 sensors-22-06644-f013:**
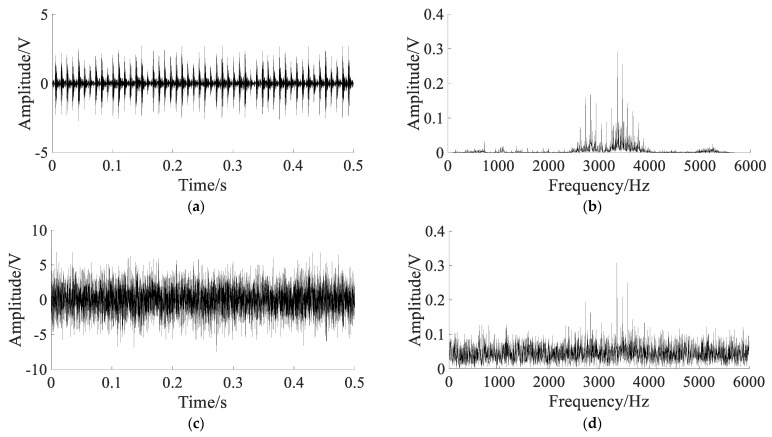
The bearing outer ring fault signals: (**a**) original signal; (**b**) spectrum of the original signal; (**c**) noisy signal; (**d**) spectrum of the noisy signal.

**Figure 14 sensors-22-06644-f014:**
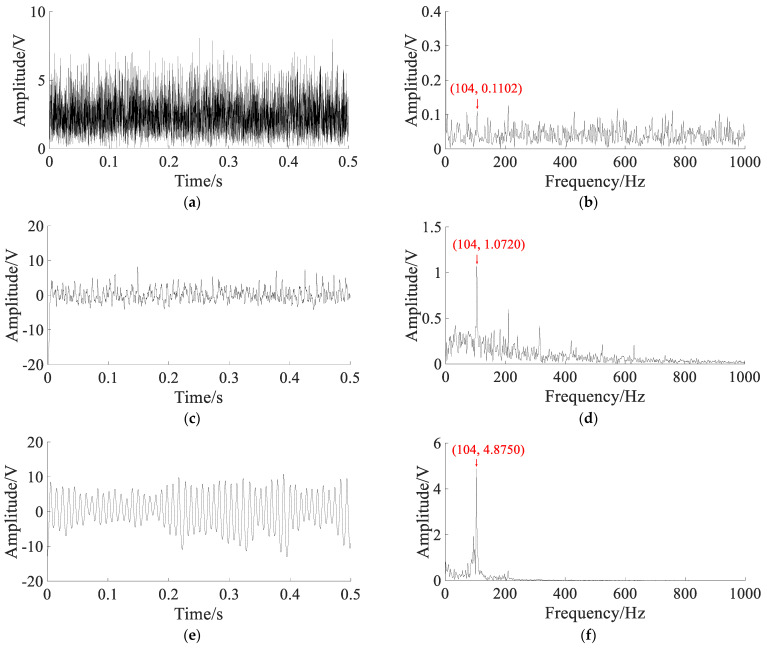
The results of different methods: (**a**) Hilbert transform envelope demodulation signal; (**b**) Hilbert envelope spectrum; (**c**) output signal of SGST adaptive SR: a=0.0245, b=0.0011, m=3000, γ=0.6828; (**d**) output signal spectrum of SGST adaptive SR; (**e**) output signal of SAFRM adaptive SR: R=3000, γ=0.0118, ε=0.2515; (**f**) output signal spectrum of SAFRM adaptive SR.

**Figure 15 sensors-22-06644-f015:**
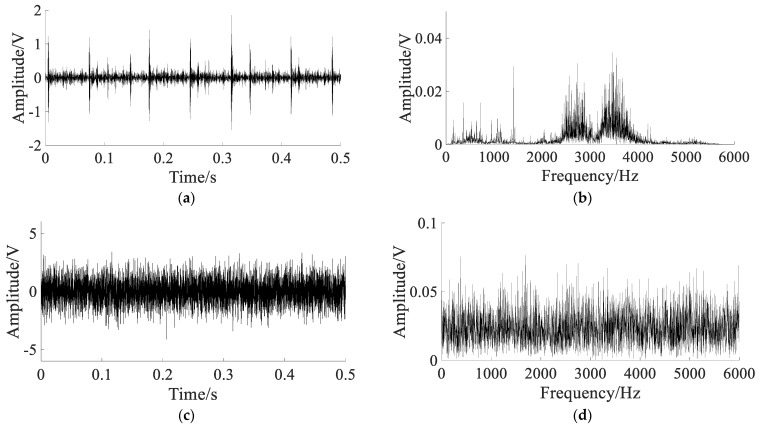
The bearing inner ring fault signals: (**a**) original signal; (**b**) spectrum of the original signal; (**c**) noisy signal; (**d**) spectrum of the noisy signal.

**Figure 16 sensors-22-06644-f016:**
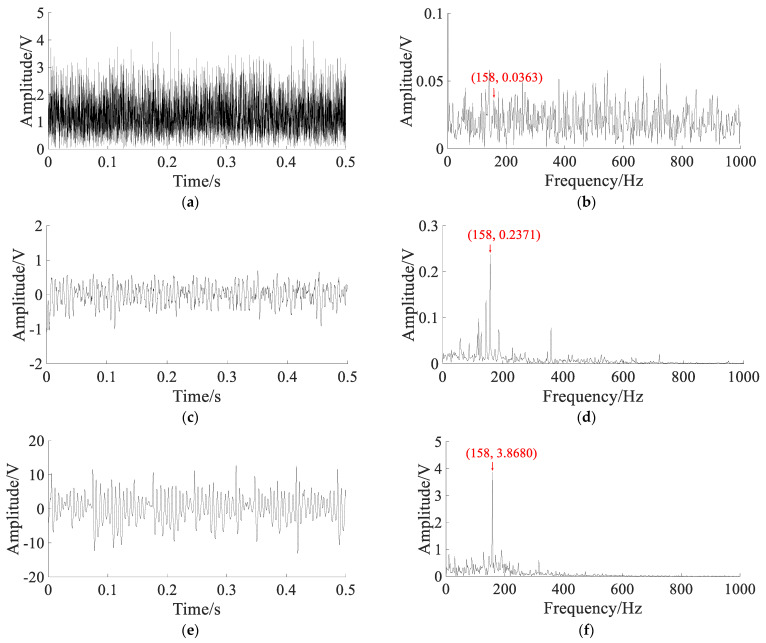
The results of analyzing the bearing inner ring fault signal of the CWRU using different methods: (**a**) Hilbert transform envelope demodulation signal; (**b**) Hilbert envelope spectrum; (**c**) output signal of SGST adaptive SR: a=0.1952, b=0.6218, m=3500, γ=0.0362; (**d**) output signal spectrum of SGST adaptive SR; (**e**) output signal of SAFRM adaptive SR: R=3500, γ=0.0927, ε=0.2618; (**f**) output signal spectrum of SAFRM adaptive SR.

**Table 1 sensors-22-06644-t001:** Index values under different noise intensities (six groups of data are selected).

Count	Noise Intensity	*PSK*	*CC*	*SSIM*	*RMSE*	*ApEn*	*SNR*
1	0.05	913.5260	0.9151	0.9954	0.3182	0.5156	22.9475
2	0.10	827.3234	0.8404	0.9924	0.4569	0.7350	16.9511
3	0.15	756.7657	0.7851	0.9843	0.5524	0.8950	13.4987
…	…	…	…	…	…	…	…
198	9.90	4.7536	0.1235	0.0059	4.4385	1.8435	−25.6313
199	9.95	4.6039	0.1219	0.0057	4.4473	1.8467	−27.1864
200	10	4.3926	0.1215	0.0056	4.4978	1.8483	−28.1692

**Table 2 sensors-22-06644-t002:** The results of the three methods for simulated signals.

	Amplitude/V	CEI of the Output Signal	Amplitude Multiplier (Compared to the Noisy Signal)	Reduced CEI (Compared to the Noisy Signal)
Noisy signal	0.0164	0.6243	1	0
Hilbert envelope demodulation	0.0456	0.3777	2.78	0.2466
SGST adaptive SR	0.5154	0.1826	31.43	0.4417
Proposed method	2.2560	0.0852	137.56	0.5391

**Table 3 sensors-22-06644-t003:** The parameters of the faulty bearing.

Fault Location	Fault Diameter (Inches)	Motor Load (HP)	Approximate Motor Speed (rpm)	Fault Characteristic Frequency (Hz)
Outer ring	0.007	2	1750	104.6
Inner ring	0.014	2	1750	157.9

**Table 4 sensors-22-06644-t004:** The results of the three methods for fault signals.

		Amplitude/V	CEI of the Output Signal	Amplitude Multiplier (Compared to the Noisy Signal)	Reduced CEI (Compared to the Noisy Signal)
Outer ring	Noisy signal	0.0706	0.7309	1	0
	Hilbert envelope demodulation	0.1102	0.5323	1.56	0.1986
	SGST adaptive SR	1.0720	0.2555	15.18	0.4754
	Method of this paper	4.875	0.0820	69.05	0.6489
Inner ring	Noisy signal	0.0249	0.6142	1	0
	Hilbert envelope demodulation	0.0363	0.4076	1.46	0.2066
	SGST adaptive SR	0.2371	0.1743	9.52	0.4399
	Method of this paper	3.8680	0.1297	155.34	0.4845

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
