# Peer review of "Weak Fault Feature Extraction Method Based on Improved Stochastic Resonance"

_sensors, 2022, doi:10.3390/s22176644_

Round 1

Reviewer 1 Report

1. In Section 2, how is Eq. (2) derived? This is the basis for the research content of this paper, and a detailed derivation should be given. Similarly, Eq. (5) is the output SNR for an overdamped SR system, but not for an underdamped SR system. The output SNR corresponding to Eq. (1) should be provided in the paper. Therefore, some conclusions based on Eq. (5) are also unconvincing in this case.

2. What does the variable e in Eq. (13) represent?

3. Only the parameters γ and ε are optimized in this paper, and the parameter R only gives the value range; in this case, how to determine the optimal value of R?

4. The method for determining the index CEI is very similar to the method in the literature "Novel synthetic index-based adaptive stochastic resonance method and its application in bearing fault diagnosis", which fails to reflect the highlights of the research work in this paper.

Reviewer 2 Report

comments are attached.

Round 2

Reviewer 1 Report

I have no other comments.

Reviewer 2 Report

The manuscript was modified according to questions.